# Prevalence, risk factors, and viral load quantification of HBV infection among hospital patients: A cross-sectional investigation in Mogadishu, Somalia

**Bashiru Garba** [1,2,3]*, **Amino Adan Mohamud**[4], **Fartuun Hassan Mohamed**[4], **Ayan Aden Moussa**[4], **Yusuf Yakubu**[5], **Najib Isse Dirie**[1,6], **Aya Muktar Abdulatif**[7], **Jamal Hassan Mohamoud**[1,2], **Abdirasak Sharif Ali**[4], **Abdurrahman Hassan Jibril**[3]

**1** SIMAD Institute for Global Health (SIGHt), SIMAD University, Mogadishu, Somalia, **2** Department of Public Health, Faculty of Medicine & Health Sciences, SIMAD University, Mogadishu, Somalia, **3** Department of Veterinary Public Health & Preventive Medicine, Faculty of Veterinary Medicine, Usmanu Danfodiyo University, Sokoto, Nigeria, **4** Department of Microbiology and Laboratory Sciences, Faculty of Medicine & Health Sciences, SIMAD University, Mogadishu, Somalia, **5** Department of Public Health and Preventive Medicine, School of Medicine, St. George's University, True Blue, Grenada, **6** Urology Department, Faculty of Medicine and Health Sciences, SIMAD University, Mogadishu, Somalia, **7** Department of Medical Laboratory Sciences, Faculty of Medical Technology, University of Tripoli, Tripoli, Libya

* garba.bashiru@simad.edu.so

## Abstract

### Background

Hepatitis B virus (HBV) remains a major public health challenge in sub-Saharan Africa. Somalia, with its fragile healthcare system and high-risk population, lacks up to date data on HBV epidemiology. This study assessed HBV prevalence, associated risk factors, and viral load among hospital patients in Mogadishu, Somalia.

### Methods

A cross-sectional survey was conducted in four major hospitals in Mogadishu. Socio-demographic and clinical risk data were collected using structured questionnaires. Blood samples from 270 participants were screened for HBsAg using a rapid test, with positives confirmed and quantified for HBV DNA by real-time PCR. Bivariate and multivariate analyses identified independent risk factors.

### Results

HBsAg prevalence was 11.5% (n = 31), but only 2.2% of all participants (6/270; 19.4% of HBsAg-positives) had detectable HBV DNA. The strongest independent predictors were family history of HBV (AOR: 5.08, 95% CI 1.90–13.60) and history of tooth extraction (AOR: 3.18, 95% CI 1.24–8.12). Socio-economic factors, facility variations, and low vaccination were also observed. Classical behavioral risk factors were

**Data availability statement:** All relevant data are within the manuscript.

**Funding:** The author(s) received no specific funding for this work.

**Competing interests:** Authors declare they have no competing interest.

infrequent and not significant. Among PCR positive cases, only one presented with a high viral load (>20,000 IU/mL).

## Conclusion

HBV poses a substantial, ongoing threat to the Somali population, especially in hospital-based settings. Household transmission and unsafe medical procedures are the most significant risk factors. Improving infection control, expanding vaccination, and prioritizing molecular diagnostics will be critical to effective HBV control in Somalia.

## Introduction

Hepatitis B virus infection remains a major public health challenge and a leading cause of morbidity and mortality globally. Despite the availability of vaccines, the diseases have continued to persist particularly in low and middle-income countries [1,2]. Available reports indicate that an estimated 296 million people live with chronic hepatitis B infection, leading to an annual mortality rate of 820,000 mainly due to complications including liver cirrhosis and hepatocellular carcinoma [3,4]. The burden of HBV is disproportionately high in the WHO African Region, which has a high prevalence of chronic infection, with an estimated 81 million people affected [4]. The virus is transmitted through contact with infected blood or body fluids such as semen and vaginal secretions, with common routes including mother-to-child transmission, unsafe medical injections, unprotected sexual contact, and other percutaneous exposures [5].

In contrast to many other African nations, comprehensive and recent data on the sero-prevalence of HBV in Somalia is severely limited. The country has faced decades of conflict and fragility, which have significantly disrupted its healthcare system, including routine immunization programs, disease surveillance, and blood safety protocols [6,7]. Conflict and humanitarian crises as experienced in Somalia create conditions, like displacement, disrupted healthcare, unsafe medical practices, and exposure to sexual violence that significantly increase vulnerability to blood-borne and sexually transmitted infections [8]. While some older studies and regional models suggest a high-intermediate endemicity for HBV in Somalia, there is a critical lack of robust studies that employ modern diagnostic techniques to quantify the current burden and transmissibility of the virus. Understanding the local epidemiology is further hampered by a general lack of information on the knowledge, attitudes, and practices of both the population and healthcare workers regarding HBV prevention and transmission [7,9].

Despite the humanitarian situation, Somalia defies the odds to become the 71st GAVI-eligible country to introduce pentavalent vaccine, that combines protection against diphtheria, tetanus, pertussis, Haemophilus influenzae type b, and HBV for children under one year of age [10]. Notwithsanding this policy, routine coverage with the three-dose pentavalent series has remained suboptimal and variable across regions, often below the national target of 90%, reflecting persistent challenges

related to insecurity, health system fragility, and vaccine stockouts [11,12]. As a result, many adolescents and adults in Mogadishu were born before the introduction of HBV-containing vaccines or may have missed infant doses, contributing to a large cohort that remains susceptible to infection and justifying the need to better characterise HBV epidemiology in hospital-attending populations [13,14].

Hospital patients represent a crucial sentinel population for understanding the epidemiology of infectious diseases within a community. They often reflect the broader population's health status and can provide early warnings of high disease burdens [15]. For HBV, determining the prevalence alone is insufficient for guiding clinical and public health interventions. Identifying the specific sociodemographic and behavioral risk factors including medical history are essential for targeted prevention strategies [16,17]. Furthermore, viral load quantification in HBsAg-positive patients is essential for guiding treatment decisions, monitoring disease progression, and determining prognosis, enabling clinicians to assess infectivity, identify candidates for antiviral therapy, and evaluate response to treatment [18–20].

Therefore, we undertook a cross-sectional study across four major hospitals in Mogadishu, Somalia in order to address a critical knowledge gap and generate evidence that will support national HBV control efforts. Our approach integrated serological screening, surveys on knowledge, attitudes, and practices, and virological quantification to provide a thorough assessment of hepatitis B in Mogadishu, Somalia.

## Methodology

### Ethics approval and consent to participate

The study was approved by the ethical committee of SIMAD University Institutional Review Board (Ref: 0002/SU/SGHt-IRB-016-12-2024). The study was conducted in accordance with the Declaration of Helsinki. Informed verbal consent was obtained from all participants prior to data collection. The consent process was conducted in Somali, ensuring comprehension of study objectives, procedures, and rights to withdraw at any time, as recommended by the ethical committee.

### Inclusivity in global research

Additional information regarding the ethical, cultural, and scientific considerations specific to inclusivity in global research is included in the Supporting Information (S1 File).

### Study design, and sampling

A hospital-based cross-sectional study was conducted in four selected hospitals in Mogadishu, Somalia that provide essential medical, surgical, and pediatric services to a population heavily affected by decades of limited healthcare infrastructure, including a large population of internally displaced persons.

A total of 270 samples were collected including 15% non-response rate based on an 18.9% pooled prevalence of HBV infection previously reported in Somalia [21], a 95% confidence interval, and a 5% margin of error.

All the patients attending outpatient departments between the period 25th January 2025–16th June 2025, who gave informed consent were included in this study, regardless of any previous testing history. For each consenting participant, medical history and other risk factors data were obtained using a pre-designed questionnaire.

### Questionnaire and data collection

The questionnaire was adopted and slightly modified from a previously published study that assessed the prevalence and risk factors associated with Hepatitis B virus infection among blood donors in Mogadishu Somalia [22]. Data were collected through face-to-face interviews administered by trained research assistants. The questionnaire was initially developed in English, translated into Somali, and back – translated to English to ensure conceptual consistency. The questionnaire captured three main domains, socio-demographic characteristics: Age, sex, marital status, education level,

occupation, and residential area; knowledge, attitudes, and practices (KAP): Knowledge about HBV transmission, prevention, and vaccination; attitudes towards infected individuals; and practices related to risk factors; as well as assessment of risk factors: history of surgery, dental procedures, blood transfusion, hospitalization, multiple sexual partners, family history, and past HBV vaccination status. Importantly, history of HBV vaccination and other clinical risk factors (including prior blood transfusion, surgery, and comorbidities) was obtained through participant self-report during the interviewers, without systematic verification against clinical or vaccination records.

## Blood sample collection and serology

Approximately 5 ml of venous blood was collected aseptically from each participant into a plain tube. The sample was allowed to clot at room temperature and then centrifuged at 4000 rpm for 10 minutes to separate the serum. The separated serum was aliquoted into two cryovials; one was stored at 4°C for immediate serological testing, and the other at −80°C for subsequent molecular analysis.

All serum samples were screened for Hepatitis B surface antigen (HBsAg) using an ISO certified Advanced quality One Step HBsAg test kit (InTec Product, Inc., Xiamen, Fujian, China), following the manufacturer's instructions. The rapid test kit has a sensitivity and specificity of 100% and 99.43% respectively. The Intecasi One Step HBsAg Test is a rapid, immunochromatographic assay for qualitative detection of hepatitis B surface antigen (HBsAg) in human whole blood, serum, or plasma. The procedure involves adding a collected specimen to the test device, waiting for the sample to migrate along the membrane, and reading results after about 15 minutes: two visible lines indicate a positive result (test and control), one line in the control region indicates negative, and no lines or a missing control line indicates an invalid test.

## Molecular methods for viral load quantitation

All serum samples that tested positive for HBsAg were advanced to molecular analysis for the detection and quantification of HBV DNA. Viral nucleic acid was extracted from 200 μL of serum using the GeneProof Pathogen Free DNA Isolation Kit (GeneProof, Brno, Czech Republic), following the manufacturer's instructions [23]. This process involved lysing the viral particles, binding the DNA to a silica membrane, washing away impurities, and finally eluting the purified DNA in a 50 μL elution buffer. The extracted DNA was stored at −20°C until the amplification step.

The quantification of HBV DNA was performed using the GeneProof HBV Quantitative RT-PCR Kit, a CE-IVD-certified assay. This real-time PCR test targets a conserved segment of the HBV surface (S) gene. The amplification was carried out on a LineGene 9600 (Bioer, China) thermal cycler in a 25 μL reaction volume, containing 20 μL of Master Mix and 5 μL of the extracted DNA. The thermal cycling protocol involved an initial activation step at 95°C for 10 minutes, followed by 45 cycles of denaturation at 95°C for 10 seconds and a combined annealing/extension phase at 60°C for 60 seconds. Each run included the kit's internal control to monitor for inhibition and extraction efficiency, as well as positive and negative controls to ensure validity. The assay has a broad dynamic range, capable of quantifying viral load from 20 IU/mL to $1 \times 10^8$ IU/mL. Based on the quantified viral load in IU/mL, HBsAg-positive patients were categorized into low (<$2 \times 10^3$ IU/mL), moderate ($2 \times 10^3$ - $2 \times 10^5$ IU/mL), or high (>$2 \times 10^5$ IU/mL) viremia groups.

## Statistical analysis

Data were exported into Microsoft Excel and analysed using SPSS version 26. Descriptive statistics was used to summarise participants' demographic characteristics, knowledge, attitudes, and practices toward hepatitis B virus (HBV), as well as the prevalence of HBsAg positivity across the four study hospitals. Continuous variables such as age were presented as means and standard deviations, while categorical variables such as sex, education, or vaccination status were expressed as frequencies and percentages.

Knowledge, attitude, and practice scores were computed from questionnaire items, assigning one point for each correct or desirable response. Scores were converted to percentages and categorised using Bloom's cut-off points as good (≥ 80%), moderate (60–79%), or poor (<60%) [24–26]. Cronbach's alpha was calculated to assess internal consistency of KAP scales. Associations between KAP levels and HBV infection were analysed both categorically and as continuous variables to test for trends.

To explore factors associated with HBV infection, bivariate analyses were performed using the chi-square for categorical variables and the t-test for continuous variables, as appropriate. Variables with a p-value <0.20 in the bivariate analysis and those identified a priori (age, sex, hospital, vaccination status) were entered into a multivariable logistic regression model to determine independent predictors of HBsAg positivity. Adjusted odds ratios (AORs) with 95% confidence intervals (CIs) were reported, and a p-value <0.05 was considered statistically significant.

For HBsAg-positive samples, real-time PCR results were analysed to determine the proportion of patients with detectable HBV DNA and to quantify viral load. Viral loads were expressed in IU/mL, log-transformed to achieve normality, and classified into three categories: low ($<2 \times 10^3$ IU/mL), moderate ($2 \times 10^3$ - $2 \times 10^5$ IU/mL), and high ($>2 \times 10^5$ IU/mL). Linear regression was used to assess relationships between $\log_{10}$ viral load and selected clinical or behavioural factors, while logistic regression examined associations with detectability of HBV DNA.

## Results

Participants were evenly distributed across the four hospitals, mostly young adults (mean age 33 years) and female being the majority (67.8%). The majority were married (64%), and nearly half had no formal education. Although most respondents were employed (66%), their income levels were low, with over 90% earning ≤ 500 USD monthly (Table 1).

Table 2 describes the clinical history and potential risk factors for HBV infection among the participants. The data reveals a population with a generally low prevalence of classic high-risk behaviors but a high frequency of certain medical exposures.

Most participants (84.1%) reported no family history of HBV, and only 16% had ever received HBV vaccination, indicating low immunization coverage among hospital attendees. Results for other risk factors for HBV infection showed that cigarette smoking (4%), sharing of sharp objects (4%), and having multiple sexual partners (4%) were relatively uncommon behaviors.

An almost equal number of the participants reported undergoing surgical operations (21.5%) and received blood transfusions (20.7%), while 47% reported tooth extraction.

Regarding comorbidities, the study found that 15.2% of the participants had diabetes while 14.1% indicated they have chronic liver disease, this was self-reported and not further sub-classified by underlying cause.

Out of the 270 blood samples screened for hepatitis B surface antigen (HBsAg) using a rapid test kit, 31 samples (11.5%) were found to be positive, while 239 (88.5%) tested negative. All 31 rapid test-positive samples were subsequently subjected to confirmatory molecular testing by real-time PCR (RT-PCR) to verify HBV DNA presence and quantify viral load (Fig 1).

Out of the 270 blood samples screened for HBsAg, 31 (11.5%) were positive and all 31 HBsAg-positive samples were subsequently tested by real-time PCR. Among these, 6 patients (19.4% of HBsAg-positives; 2.2% of the total sample) had detectable HBV DNA, indicating active viral replication, while 25 HBsAg-positive patients had HBV DNA levels below the assay's limit of detection. HBV DNA testing was not performed on HBsAg-negative samples, so the presence of occult HBV infection in this group cannot be excluded. Viral load quantification showed that among the six PCR-positive cases, only one had a viral load exceeding 20,000 IU/mL, indicating high-level viral replication consistent with clinically significant HBV activity.

The real-time PCR amplification plot (Fig 2-A) demonstrates sigmoidal fluorescence curves corresponding to six HBV DNA-positive samples. Each curve represents a distinct amplification event, with the exponential phase occurring between

**Table 1. Sociodemographic characteristics of patients enrolled for HBV infection screening across four public hospitals in Mogadishu, Somalia (N = 270).**

| Variable | Category | Counts | Percentage of Total |
|---|---|---|---|
| Hospital | | | |
| | Laadnan | 67 | 24.8% |
| | Demartini | 67 | 24.8% |
| | SOS | 70 | 25.9% |
| | Sumait | 66 | 24.4% |
| Age | | | |
| | Less than 18 years | 14 | 5.2% |
| | 18-30 years | 132 | 48.9% |
| | 31-40 years | 67 | 24.8% |
| | More than 40 years | 57 | 21.1% |
| Mean ± SD 33.1 ± 12.9 | | | |
| Sex | | | |
| | Male | 87 | 32.2% |
| | Female | 183 | 67.8% |
| Marital status | | | |
| | Single | 77 | 28.5% |
| | Married | 174 | 64.4% |
| | Widowed/Divorced | 19 | 7.0% |
| Educational qualification | | | |
| | No education | 130 | 48.1% |
| | Primary | 13 | 4.8% |
| | Secondary | 37 | 13.7% |
| | Tertiary/University | 78 | 28.9% |
| | Dugsi/Madrasa* | 12 | 4.4% |
| Occupational status | | | |
| | Employed | 177 | 65.6% |
| | Unemployed | 93 | 34.4% |
| Monthly income | | | |
| | Less than 100 USD | 114 | 42.2% |
| | 100-500 USD | 133 | 49.3% |
| | More than 500 USD | 23 | 8.5% |
| Number of marriages | | | |
| | Not applicable | 75 | 27.8% |
| | Once | 164 | 60.7% |
| | Twice | 21 | 7.8% |
| | Three times | 6 | 2.2% |
| | More than 3 times | 4 | 1.5% |

Note: *Dugsi/Madrasa means Islamic religious schools.

cycle 25 and 40, confirming successful detection of HBV DNA in these specimens. The earlier cycle threshold (Ct) values observed in one sample indicate a higher viral copy number, consistent with the quantitative output showing a viral load >20,000 IU/mL.

**Table 2. Clinical history, behavioral risk factors, and HBV vaccination status of hospital patients in Mogadishu, Somalia (N = 270).**

| Variable | Category | Counts | % of Total |
|---|---|---|---|
| History of HBV in the family | | | |
| | No | 227 | 84.1% |
| | Yes | 43 | 15.9% |
| Cigarette smoking | | | |
| | No | 258 | 95.6% |
| | Yes | 12 | 4.4% |
| Have you received blood donations before? | | | |
| | No | 214 | 79.3% |
| | Yes | 56 | 20.7% |
| History of HBV vaccination | | | |
| | No | 226 | 83.7% |
| | Yes | 44 | 16.3% |
| History of sharing needles, nail cutter, razor blades | | | |
| | No | 258 | 95.6% |
| | Yes | 12 | 4.4% |
| History of tooth extraction | | | |
| | No | 143 | 53.0% |
| | Yes | 127 | 47.0% |
| History of surgical operation | | | |
| | No | 212 | 78.5% |
| | Yes | 58 | 21.5% |
| History of blood or organ donation | | | |
| | No | 260 | 96.3% |
| | Yes | 10 | 3.7% |
| Multiple sexual partners | | | |
| | No | 259 | 95.9% |
| | Yes | 11 | 4.1% |
| History of chronic liver disease | | | |
| | No | 232 | 85.9% |
| | Yes | 38 | 14.1% |
| History of Sexually Transmitted Infections (STIs) | | | |
| | No | 228 | 84.4% |
| | Yes | 42 | 15.6% |
| History of diabetes | | | |
| | No | 229 | 84.8% |
| | Yes | 41 | 15.2% |

* Statistically significant (p < 0.05)

The accompanying standard curve (Fig 2-B) shows a strong linear relationship between Ct values and $\log_{10}$ DNA concentration ($R^2 = 0.971$), demonstrating high assay precision and reproducibility. The slope of –3.17 and efficiency of 106% is the acceptable limits for diagnostic real-time PCR assays, confirming that the amplification reaction was both efficient and accurate across dilutions.

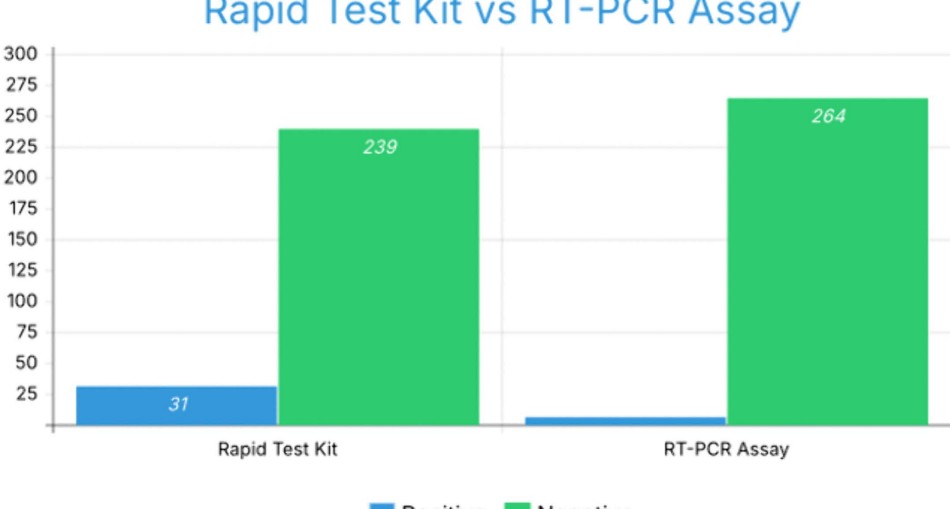

**Fig 1. Distribution of positive and negative results by serology and RT-PCR.**

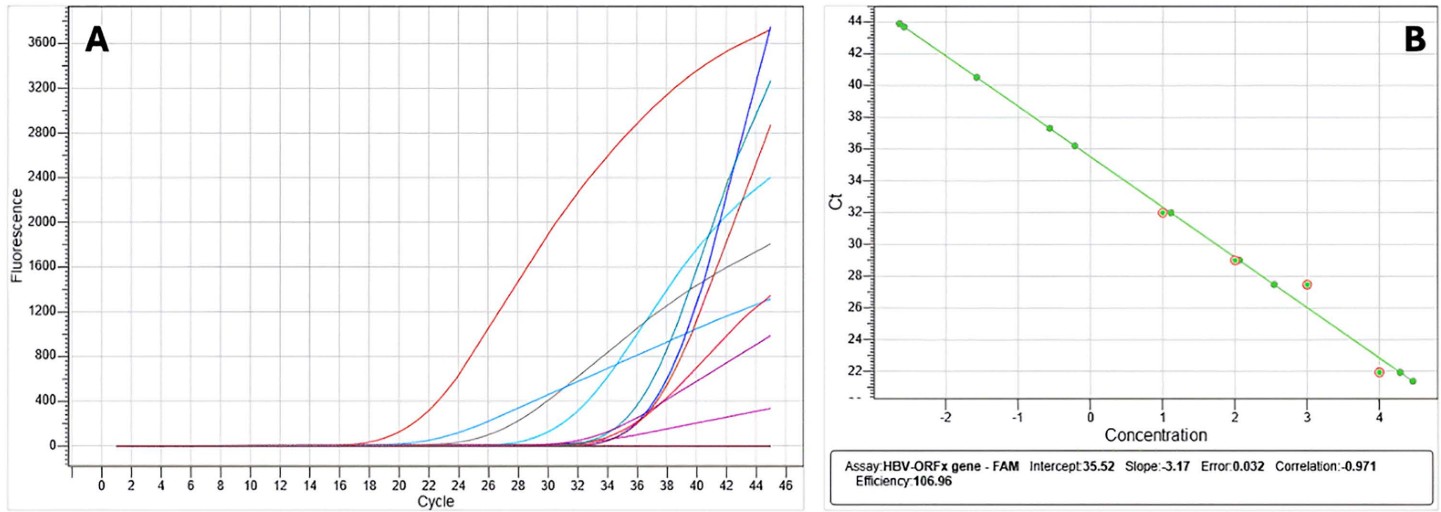

**Fig 2. Real-time PCR assay.** A: Amplification curve showing in real time the tests and controls according to cycle thresholds (CT). B: Standard curve showing Ct values versus log DNA concentration, used for accurate HBV DNA quantification in RT-PCR.

The bivariate analysis conducted using chi-square identified several sociodemographic factors significantly associated with Hepatitis B surface antigen (HBsAg) seropositivity including, hospital, marital status, monthly income, and number of marriages (p < 0.05) (Table 3).

HBsAg positivity varied across hospitals (p = 0.007), with cases reported in Laadnan, Demartino, and SOS Hospitals, but none in Sumait Hospital, suggesting possible facility-level differences in patient exposure or screening coverage. HBV infection was significantly higher among married participants (14.9%) compared to singles (1.3%), while participants

**Table 3. Bivariate association between sociodemographic factors and HBsAg status among participants (n = 270).**

| Variable | Category | HbsAg Negative (n = 239) | HbsAg-Positive (n = 31) | Total (N = 270) | χ² Value | p-value |
|---|---|---|---|---|---|---|
| Hospital | | | | | 12.2 | 0.007* |
| | Laadnan | 57 | 10 | 67 | | |
| | Demartini | 55 | 12 | 67 | | |
| | SOS | 61 | 9 | 70 | | |
| | Sumait | 66 | 0 | 66 | | |
| RT-PCR result | | | | | 47.3 | 0.001* |
| | Negative | 239 | 25 | 264 | | |
| | Positive | 0 | 6 | 6 | | |
| Age group | | | | | 2.69 | 0.442 |
| | < 18 years | 12 | 2 | 14 | | |
| | 18-30 years | 121 | 11 | 132 | | |
| | 31-40 years | 58 | 9 | 67 | | |
| | > 40 years | 48 | 9 | 57 | | |
| Sex | | | | | 0.163 | 0.686 |
| | Male | 78 | 9 | 87 | | |
| | Female | 161 | 22 | 183 | | |
| Marital status | | | | | 11.6 | 0.003* |
| | Single | 76 | 1 | 77 | | |
| | Married | 148 | 26 | 174 | | |
| | Widowed/Divorced | 15 | 4 | 19 | | |
| Education | | | | | 2.15 | 0.709 |
| | No education | 112 | 18 | 130 | | |
| | Primary | 11 | 2 | 13 | | |
| | Secondary | 33 | 4 | 37 | | |
| | Tertiary/University | 72 | 6 | 78 | | |
| | Dugsi/Madrasa | 11 | 1 | 12 | | |
| Occupational status | | | | | 0.282 | 0.595 |
| | Employed | 158 | 19 | 177 | | |
| | Unemployed | 81 | 12 | 93 | | |
| Monthly income | | | | | 7.74 | 0.021* |
| | < 100 USD | 108 | 6 | 114 | | |
| | 100-500 USD | 111 | 22 | 133 | | |
| | > 500 USD | 20 | 3 | 23 | | |
| Number of marriages | | | | | 10.7 | 0.031* |
| | Not applicable | 71 | 4 | 75 | | |
| | Once | 140 | 24 | 164 | | |
| | Twice | 21 | 0 | 21 | | |
| | Three times | 4 | 2 | 6 | | |
| | > 3 times | 3 | 1 | 4 | | |

earning 100–500 USD/month had the highest infection rate (16.5%). A significant association was also noted with number of marriages (p = 0.031), where those married once showed higher infection rates than other groups.

No significant relationship was found between HBV infection and age, sex, education, or occupation (p > 0.05). As expected, RT-PCR results correlated strongly with HBsAg positivity (p < 0.001), confirming serological reliability.

This analysis reveals that a family history of HBV and specific medical procedures are the most significant risk factors associated with Hepatitis B infection in this population, while classic behavioral risks were less common and not statistically significant.

Participants with a family history of HBV were substantially more likely to be HBsAg positive (14/43; 32.6%) than those without such history (17/227; 7.5%) (χ² = 22.4, p < 0.001), indicating strong evidence for intra-household or vertical transmission.

A significant relationship was also observed with tooth extraction history (χ² = 10.4, p = 0.001), where infection was higher among those who had undergone extractions (23/127; 18.1%) compared with those who had not (8/143; 5.6%) (Table 4).

Similarly, participants reporting a history of STIs were more likely to be HBsAg positive (9/42; 21.4%) than those without (22/228; 9.6%) (χ² = 4.84, p = 0.028). however, other factors including cigarette smoking, blood transfusion, surgical history, organ donation, sharing of needles or sharp objects, and chronic diseases such as diabetes or liver disease showed no statistically significant association with HBV infection (p > 0.05). Although transfusion history approached significance (p = 0.093), it did not reach the conventional threshold, suggesting only a weak relationship in this sample.

The multivariate logistic regression analysis identifies independent predictors of HBsAg positivity among study participants. Only two variables were found to be statistically significant (p < 0.05), meaning their associations with HBV infection are unlikely due to chance after adjusting for confounding factors in the model (Table 5).

Participants with a family history of HBV were about five times more likely to test positive for HBsAg compared with those without such history (AOR = 5.08; 95% CI: 1.90–13.60; p = 0.001). Likewise, individuals with a history of tooth extraction had more than threefold higher odds of HBV infection (AOR = 3.18; 95% CI: 1.24–8.12; p = 0.016) compared with those without such history.

Other variables, including cigarette smoking, blood transfusion, HBV vaccination, surgical history, sharing of sharp instruments, multiple sexual partners, and chronic diseases (STIs, diabetes, or liver disease), were not significantly associated with HBsAg positivity (p > 0.05). Although some (such as blood transfusion and surgical procedures) showed elevated odds ratios (> 1), these did not reach statistical significance, possibly due to the modest sample size or low event counts in specific categories.

Marital history did not show a consistent or significant pattern; participants married once were used as the reference group, and none of the alternative categories differed significantly.

## Discussion

This hospital-based cross-sectional study provides important insights into the current epidemiology of Hepatitis B Virus infection in Mogadishu, Somalia, a setting characterized by a fragile health system and a scarcity of recent robust data. The integrated approach used, combining serology, molecular confirmation, and risk factor analysis, reveals three key findings: a substantial seroprevalence of HBV, a much lower rate of active viremia, and familial and iatrogenic exposures as the dominant independent risk factors.

The seroprevalence of HBsAg among hospital patients was found to be 11.5%. This figure aligns with the global estimation where African and Western Pacific regions are classified as areas of high-intermediate to high endemicity, with hepatitis B prevalence ranging from about 5% to over 8%, and in several countries, estimates exceed 15% [27]. Important to highlight that although the high prevalence recorded in this study (11.5%) reflects a hospital-based population and may not represent the general community, it nonetheless indicates a high burden of infection and potential ongoing transmission within the population, which is a course for concern [28]. This result is consistent with previous Somali studies, where the pooled prevalence was 18.9%, with rates of 13.3% among pregnant women and 9.7% among blood donors in Mogadishu [21,22,29]. Also, similar findings have been reported in neighboring countries also experiencing humanitarian crisis including Sudan and South Sudan where a prevalence of 6.8% and 26% were reported in central Sudan and southern

**Table 4. Bivariate analysis of clinical, familial, and behavioral risk factors associated with HBV seropositivity.**

| Variable | Category | HBsAg-Negative (n = 239) | HBsAg-Positive (n = 31) | Total (N = 270) | X²-value | p-value |
|---|---|---|---|---|---|---|
| Family history of HBV | | | | | 22.4 | < .001* |
| | No | 210 | 17 | 227 | | |
| | Yes | 29 | 14 | 43 | | |
| Cigarette smoking | | | | | 0.332 | 0.564 |
| | No | 229 | 29 | 258 | | |
| | Yes | 10 | 2 | 12 | | |
| History of blood transfusion | | | | | 2.83 | 0.093 |
| | No | 193 | 21 | 214 | | |
| | Yes | 46 | 10 | 56 | | |
| HBV vaccination history | | | | | 2.49 | 0.115 |
| | No | 197 | 29 | 226 | | |
| | Yes | 42 | 2 | 44 | | |
| Sharing needles/razors | | | | | 0.122 | 0.726 |
| | No | 228 | 30 | 258 | | |
| | Yes | 11 | 1 | 12 | | |
| History of tooth extraction | | | | | 10.4 | 0.001* |
| | No | 135 | 8 | 143 | | |
| | Yes | 104 | 23 | 127 | | |
| History of surgical operation | | | | | 0.025 | 0.874 |
| | No | 188 | 24 | 212 | | |
| | Yes | 51 | 7 | 58 | | |
| History of blood/organ donation | | | | | 0.022 | 0.881 |
| | No | 230 | 30 | 260 | | |
| | Yes | 9 | 1 | 10 | | |
| Multiple sexual partners | | | | | 0.065 | 0.800 |
| | No | 229 | 30 | 259 | | |
| | Yes | 10 | 1 | 11 | | |
| History of chronic liver disease | | | | | 2.10 | 0.148 |
| | No | 208 | 24 | 232 | | |
| | Yes | 31 | 7 | 38 | | |
| History of STIs | | | | | 4.84 | 0.028* |
| | No | 206 | 22 | 228 | | |
| | Yes | 33 | 9 | 42 | | |
| History of diabetes | | | | | 0.473 | 0.492 |
| | No | 204 | 25 | 229 | | |
| | Yes | 35 | 6 | 41 | | |

* Statistically significant (p < 0.05)

Sudan respectively [30]. However, it is significantly higher than the 4.3% national prevalence reported in neighboring Uganda, underscoring the heightened vulnerability and potential for silent transmission within the Somali population, probably exacerbated by the poor healthcare service delivery as a result of the long-standing humanitarian situation [31,32].

A key finding of our study was the assessment of active viremia among seropositive individuals. Of the 31 patients who tested positive for HBsAg by rapid test, only 6 (19.4%) had detectable HBV DNA upon confirmatory RT-PCR testing, indicating that a substantial proportion of serologically positive cases had viral loads below the assay's limit of detection

**Table 5. Independent predictors of HBV seropositivity from multivariate logistic regression analysis.**

| Predictor | Estimate | SE | Z | p | Odds ratio | 95% Confidence Interval | |
|---|---|---|---|---|---|---|---|
| | | | | | | Lower | Upper |
| Number of marriages | | | | | | | |
| Not Applicable | 0.7169 | 0.624 | 1.1485 | 0.251 | 2.048 | 0.60259 | 6.96 |
| Three times | 0.0684 | 1.12 | 0.0611 | 0.951 | 1.071 | 0.11917 | 9.62 |
| More than 3 times | 0.6849 | 2.928 | 0.2339 | 0.815 | 1.983 | 0.00638 | 616.22 |
| Once | REF | REF | REF | REF | 1 | NA | NA |
| History of HBV in the family | | | | | | | |
| Yes | 1.6254 | 0.503 | 3.2345 | 0.001* | 5.081* | 1.89743 | 13.6 |
| No | REF | REF | REF | REF | 1 | NA | NA |
| Cigarette smoking | | | | | | | |
| Yes | −1.4003 | 1.025 | −1.3666 | 0.172 | 0.247 | 0.03309 | 1.84 |
| No | REF | REF | REF | REF | 1 | NA | NA |
| Have you received blood donations before? | | | | | | | |
| Yes | −0.7066 | 0.542 | −1.3041 | 0.192 | 0.493 | 0.1706 | 1.43 |
| No | REF | REF | REF | REF | 1 | NA | NA |
| History of HBV vaccination | | | | | | | |
| Yes | 0.9636 | 0.814 | 1.1836 | 0.237 | 2.621 | 0.53145 | 12.93 |
| No | REF | REF | REF | REF | 1 | NA | NA |
| History of sharing needles, nail cutter, razor blades | | | | | | | |
| Yes | 0.4411 | 1.203 | 0.3667 | 0.714 | 1.554 | 0.14709 | 16.43 |
| No | REF | REF | REF | REF | 1 | NA | NA |
| History of tooth extraction | | | | | | | |
| Yes | 1.1566 | 0.479 | 2.4162 | 0.016* | 3.179* | 1.2441 | 8.12 |
| No | REF | REF | REF | REF | 1 | NA | NA |
| History of surgical operation | | | | | | | |
| Yes | 0.7338 | 0.585 | 1.2548 | 0.21 | 2.083 | 0.66209 | 6.55 |
| No | REF | REF | REF | REF | 1 | NA | NA |
| History of blood or organ donation | | | | | | | |
| Yes | 0.6117 | 1.222 | 0.5006 | 0.617 | 1.843 | 0.16812 | 20.21 |
| No | REF | REF | REF | REF | 1 | NA | NA |
| Multiple sexual partners | | | | | | | |
| Yes | 0.155 | 2.84 | 0.0546 | 0.956 | 1.168 | 0.00447 | 305.32 |
| No | REF | REF | REF | REF | 1 | NA | NA |
| History of chronic liver disease | | | | | | | |
| Yes | 0.1566 | 0.657 | 0.2384 | 0.812 | 1.17 | 0.32272 | 4.24 |
| No | REF | REF | REF | REF | 1 | NA | NA |
| History of Sexually Transmitted Infections (STIs) | | | | | | | |
| Yes | −0.7853 | 0.556 | −1.4127 | 0.158 | 0.456 | 0.15337 | 1.36 |
| No | REF | REF | REF | REF | 1 | NA | NA |
| History of diabetes | | | | | | | |
| Yes | −0.4497 | 0.628 | −0.7165 | 0.474 | 0.638 | 0.18637 | 2.18 |
| No | REF | REF | REF | REF | 1 | NA | NA |

Number of marriages (twice) was not included due to zero prevalence; *Statistical significance (p < 0.05).

at the time of sampling [33]. This indicates that among the infected individuals in our cohort, the vast majority (80.6%) were HBsAg-positive individuals without detectable HBV DNA at the time of testing, consistent with low- or non-replicative infection in this cohort. This distinction is clinically and public health critical. Although nucleic acid amplification tests are inherently more sensitive than serological assays for detecting active viral replication, commercial quantitative PCR platforms still have defined lower limits of detection and may fail to identify very low-level or fluctuating viremia [34,35]. The discordance observed between HBsAg positivity and undetectable HBV DNA in our cohort is therefore likely to reflect a predominance of inactive or low-replicative infection, and possibly resolving infection, rather than poor performance of the molecular assay. Similar patterns of HBsAg positivity with undetectable or intermittently detectable HBV DNA have been reported among inactive carriers and patients with low-replicative disease in other HBV-endemic settings, underscoring the need to interpret single time-point HBV DNA results in conjunction with serological markers and clinical context [36–38]. The 11.5% seroprevalence defines the pool of infected individuals requiring long-term monitoring for liver disease, while the subset with detectable viremia identifies those who are most infectious and are potential candidates for antiviral therapy [39,40]. Future studies in Somalia employing ultra-sensitive molecular assays with lower limits of detection and longitudinal sampling would help to better characterise low-level viremia, occult infection, and their implications for transmission risk and clinical management [41–43].

Our multivariate analysis identified a family history of HBV as the strongest independent predictor of infection (AOR = 5.08). Family history of HBV infection is a major concern because it signals sustained household-level transmission and often reflects missed opportunities for prevention [22]. This finding is consistent with studies from across Africa and Asia, which consistently highlight intra-household transmission as a major route of HBV spread [34]. In the Somali context, this likely reflects a combination of vertical (mother-to-child) transmission and horizontal transmission among children and household contacts through the sharing of razors, toothbrushes, or other personal items that may carry trace amounts of blood. The high number of occupants per household typical in Mogadishu, including in IDP camps, likely facilitates this transmission [36]. Such clustering is well documented across high-burden settings, and points to the need for targeted interventions, including screening, vaccination, and education of all household members, not just the index case. In this context, integrating family-based strategies into HBV control programs could substantially reduce new infections and interrupt persistent household transmission.

Equally significant was the independent association between history of tooth extraction and HBV infection (AOR = 3.18). This strongly points to iatrogenic transmission through dental procedures as a major risk factor [37]. In settings with weak regulatory oversight and limited resources, the sterilization of dental instruments can be inadequate, turning routine dental care into a high-risk activity for blood-borne virus transmission [38]. This finding echo concerns raised in studies from other low-resource settings and highlights a critical gap in infection prevention and control within the Somali health system [40]. Therefore, strengthening infection prevention and control protocols, particularly in private and public dental clinics, must be a priority for national HBV control programs.

Contrary to findings from some other studies, classic behavioral risk factors such as multiple sexual partners, sharing of needles/razors, and cigarette smoking were not significant in our multivariate model. This suggests that in Mogadishu, familial and healthcare-associated exposures may currently be more dominant drivers of HBV transmission than sexual or percutaneous drug use pathways. The non-significant association with a history of blood transfusion, while showing a trend, may reflect some success in blood safety screening or underreporting of this history. Moreover, the absence of statistically significant association for these high-risk behaviors could be attributed to underreporting due to social desirability bias, particularly for behaviors considered culturally sensitive in Somali and similar conservative settings [42]. Moreover, horizontal and perinatal transmission routes remain the dominant modes of infection in high-endemic regions, often overshadowing sexual or parenteral exposure as primary drivers [22,43]. The finding may also reflect limited behavioral heterogeneity among the sampled hospital population, where other contextual factors such as family exposure and inadequate vaccination play a more substantial role in determining infection risk. Hence, while behavioral factors are

biologically plausible, their epidemiological signal can be muted in settings where community- and household-level transmission predominates.

The very low HBV vaccination coverage (16.3%) in our study population is a grave concern and represents a massive, missed opportunity for prevention. This is far below the WHO's target for universal immunization and leaves the vast majority of the population susceptible to infection. The counterintuitive association between vaccination and HBsAg positivity in our model is a classic example of confounding by indication, where individuals at highest risk (e.g., those with a family history) are more likely to seek vaccination. This does not indicate vaccine ineffectiveness but rather highlights that vaccination efforts are not yet reaching the broader population.

### Limitations

This study has several limitations. The use of a rapid test for initial screening, despite its reported high sensitivity and specificity, is inferior to other diagnostic assays like ELISA. However, the subsequent PCR confirmation of positives strengthens our viremia estimates. However, this study used a single time-point HBV DNA measurement with a commercial quantitative PCR assay that has a defined lower limit of detection, so very low-level or fluctuating viremia among HBsAg-positive patients may have been missed, potentially underestimating the true burden of active infection and limiting interpretation of HBsAg-positive/HBV DNA-negative cases. Similarly, HBV DNA testing was restricted to HBsAg-positive samples; thus, occult HBV infection among HBsAg-negative individuals may have been missed, potentially leading to underestimation of the true molecular prevalence of HBV infection in this population. Secondly, the hospital-based design may limit the generalizability of our findings to the general population, and the cross-sectional nature of the study precludes the establishment of causal relationships. Finally, clinical history and HBV vaccination status were self-reported and not routinely verified against medical or vaccination records, which introduces potential recall and social desirability bias and may have led to misclassification of vaccination coverage and some risk factors. Notwithstanding our findings remain highly informative, highlighting a substantial HBV burden among healthcare seekers and offering crucial insight into the epidemiological patterns of infection in a setting like Somalia, where community-level data are scarce. Furthermore, social desirability bias may have led to underreporting of sensitive behavioral risk factors.

### Conclusion

In summary, this cross-sectional study reveals a substantial seroprevalence of hepatitis B virus among hospital patients in Mogadishu, Somalia, with an 11.5% HBsAg positivity rate but a far lower proportion exhibiting active viremia. Family history of HBV and history of tooth extraction emerged as the dominant independent risk factors, reflecting persistent household-level transmission and inadequacies in infection control during medical procedures. Traditional behavioral risks such as multiple sexual partners and sharing of sharp objects were infrequent and not significant predictors in this population. The low coverage of HBV vaccination points to major gaps in preventive programming. These findings highlight the urgent need for enhanced molecular diagnostics, targeted family-based interventions, and improved infection prevention in both public and private health services, particularly dental care.

### Supporting information

**S1 File. Inclusivity in global research.**
(DOCX)

### Acknowledgments

The authors express their sincere gratitude to the directors and management teams of Laadnan, Demartini, SOS, and Sumait Hospitals in Mogadishu for their cooperation in facilitating patient recruitment and supporting sample collection throughout the study.

## Author contributions

**Conceptualization:** Bashiru Garba, Fartuun Hassan Mohamed.

**Data curation:** Bashiru Garba, Amino Adan Mohamud, Fartuun Hassan Mohamed.

**Formal analysis:** Bashiru Garba, Amino Adan Mohamud, Fartuun Hassan Mohamed, Yusuf Yakubu.

**Investigation:** Bashiru Garba, Amino Adan Mohamud, Fartuun Hassan Mohamed.

**Methodology:** Bashiru Garba, Amino Adan Mohamud, Fartuun Hassan Mohamed.

**Project administration:** Bashiru Garba.

**Supervision:** Bashiru Garba.

**Validation:** Bashiru Garba, Najib Isse Dirie, Aya Muktar Abdulatif, Jamal Hassan Mohamud, Abdirasak Sharif Ali, Abdurrahman Hassan Jibril.

**Visualization:** Ayan Aden Moussa.

**Writing – original draft:** Bashiru Garba.

**Writing – review & editing:** Bashiru Garba, Ayan Aden Moussa, Yusuf Yakubu, Najib Isse Dirie, Aya Muktar Abdulatif, Jamal Hassan Mohamud, Abdirasak Sharif Ali, Abdurrahman Hassan Jibril.

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
