## [Decision Letter · Decision Letter 0]

1 Dec 2025

Dear Dr. Garba,

Thank you for submitting your manuscript to PLOS ONE. After careful consideration, we feel that it has merit but does not fully meet PLOS ONE’s publication criteria as it currently stands. Therefore, we invite you to submit a revised version of the manuscript that addresses the points raised during the review process.

We look forward to receiving your revised manuscript.

Kind regards,

Ashraf Elbahrawy

Academic Editor

PLOS ONE

Journal Requirements:

2. Please include a complete copy of PLOS’ questionnaire on inclusivity in global research in your revised manuscript. Our policy for research in this area aims to improve transparency in the reporting of research performed outside of researchers’ own country or community. The policy applies to researchers who have travelled to a different country to conduct research, research with Indigenous populations or their lands, and research on cultural artefacts. The questionnaire can also be requested at the journal’s discretion for any other submissions, even if these conditions are not met.

Please find more information on the policy and a link to download a blank copy of the questionnaire here: https://journals.plos.org/plosone/s/best-practices-in-research-reporting.

Please upload a completed version of your questionnaire as Supporting Information when you resubmit your manuscript.

4. Please upload a new copy of Figures 1 and 2 as the detail is not clear. Please follow the link for more information:  https://journals.plos.org/plosone/s/figures

5. We note you have included a table to which you do not refer in the text of your manuscript. Please ensure that you refer to Tables 4 and 5 in your text; if accepted, production will need this reference to link the reader to the Table.

Reviewers' comments:

Reviewer's Responses to Questions

**Comments to the Author**

1. Is the manuscript technically sound, and do the data support the conclusions?

Reviewer #1: Yes

Reviewer #2: Yes

2. Has the statistical analysis been performed appropriately and rigorously?

Reviewer #1: Yes

Reviewer #2: Yes

3. Have the authors made all data underlying the findings in their manuscript fully available?

Reviewer #1: Yes

Reviewer #2: Yes

4. Is the manuscript presented in an intelligible fashion and written in standard English?

Reviewer #1: No

Reviewer #2: Yes

Reviewer #1: This was a well-designed study that showed the prevalence and molecular detection of HBV among hospital patients in Mogadishu, Somalia. The authors clearly stated the problem and the aim of the study. The methodology used was standard for this type of study. However, some issues need to be addressed:

1) HBsAg was positive in 31 patients, and HBV DNA was detected in 6 patients. It means 25 HBsAg-positive patients did not have detectable HBV DNA. Considering the high sensitivity and specificity of HBsAg rapid test kit (100% and 99.43%), it seems real-time PCR was not sensitive enough to detect low copy numbers of HBV DNA. This issue should be discussed in the discussion section.

2) Kindly clarify in the manuscript whether participants’ vaccination history was self-reported only or confirmed using clinical records. The limitations of relying on self-reported clinical history/patient recall should be adequately addressed in the manuscript where appropriate.

3) Information on vaccination in the study area has not been mentioned in the introduction section. Vaccination coverage, when did the vaccination start, which age group is targeted, these are missing in the introduction section. These inform the readers about the situation in the study area and support why the study was conducted.

4) The limitations of the study are not sufficiently discussed.

5) I would recommend a linguistic revision of the manuscript by a native speaker as some parts do not read very well.

Reviewer #2: This study looks at the Prevalence, risk factors, and viral load quantification of HBV infection among hospital patients: A cross-sectional investigation in Mogadishu, Somalia. It is well written but I have a few concerns for the authors to address.

1. 60…. Kindly clarify the body fluids through which transmission can occur since is not all body fluids

2. 214….. Kindly clarify what Dugsi/Madrasa is

3. 229…… 14.1% indicated they have chronic liver disease, what type of chronic liver disease. Does that mean some of the patients were aware of their hepatitis B status? Did you exclude patient who have screen for HBV already?

4. 245 – 246…….In the methods, you subjected only the 31 samples to RT-PCR test, so you can claim those you did not do RT-PCR were negative. Therefore, this statement ‘’The RT-PCR results showed that 6 samples (2.2%) were confirmed positive for HBV DNA, indicating active viral replication, while 264 samples (97.8%) were negative’’ cannot be true. What about if some of the samples you did not do PCR for have occult hepatitis B infection

5. 371- 378 ….that among the infected individuals in our cohort, the vast majority (80.6%) were likely 371 inactive carriers with no detectable viral replication….. kindly replaced inactive carriers with a correct terminology

**Do you want your identity to be public for this peer review?** For information about this choice, including consent withdrawal, please see our Privacy Policy

Reviewer #1: **Yes**:  Fatemeh Farshadpour

Reviewer #2: No

---

## [Author Response · Author response to Decision Letter 1]

16 Dec 2025

Response to editorial and reviewer comments

PONE-D-25-59057: Prevalence, risk factors, and viral load quantification of HBV infection among hospital patients: A cross-sectional investigation in Mogadishu, Somalia

We would like to sincerely thank the Editor and Reviewers for their careful evaluation of our manuscript and for the insightful comments and suggestions provided. Their constructive feedback has helped us to clarify key methodological and interpretive aspects of the study, and we fully appreciate and share many of the concerns and priorities they have highlighted in relation to hepatitis B prevention and control in Somalia.

Editorial review/Journal Requirements:

Response: The manuscript has been edited to conform with the journal guidelines

2. Please include a complete copy of PLOS’ questionnaire on inclusivity in global research in your revised manuscript.

Response: The questionnaire on inclusivity in global research has been filled and submitted as supplimentary document.

3. PLOS requires an ORCID iD for the corresponding author in Editorial Manager on papers submitted after December 6th, 2016. Please ensure that you have an ORCID iD and that it is validated in Editorial Manager.

Response: I have added my ORCID to the submission system

4. Please upload a new copy of Figures 1 and 2 as the detail is not clear.

Response: An improved figure has been submitted as attachments.

5. We note you have included a table to which you do not refer in the text of your manuscript. Please ensure that you refer to Tables 4 and 5 in your text; if accepted, production will need this reference to link the reader to the Table.

Response: Both Tables 4 and 5 has been cited accordingly.

Reviewer 1

This was a well-designed study that showed the prevalence and molecular detection of HBV among hospital patients in Mogadishu, Somalia. The authors clearly stated the problem and the aim of the study. The methodology used was standard for this type of study. However, some issues need to be addressed:

1. HBsAg was positive in 31 patients, and HBV DNA was detected in 6 patients. It means 25 HBsAg-positive patients did not have detectable HBV DNA. Considering the high sensitivity and specificity of HBsAg rapid test kit (100% and 99.43%), it seems real-time PCR was not sensitive enough to detect low copy numbers of HBV DNA. This issue should be discussed in the discussion section.

Response: Thank you for this critical observation. We have now expanded the Discussion to explain that this pattern likely reflects low level or inactive HBV infection with viral loads below the PCR assay’s limit of detection, rather than suboptimal PCR assay performance, and we have supported this with additional references. We also acknowledge this as a limitation of our virological assessment and recommend future studies using more sensitive molecular assays and longitudinal sampling to better characterise low level viremia in this setting (line 387-403).

2. Kindly clarify in the manuscript whether participants’ vaccination history was self-reported only or confirmed using clinical records. The limitations of relying on self-reported clinical history/patient recall should be adequately addressed in the manuscript where appropriate.

Response: Thank you once again, we appreciate this comment. We have clarified in the Methods and Limitation that vaccination history and other clinical risk factors were collected by self report only (line 143-146; line 484-487).

3. Information on vaccination in the study area has not been mentioned in the introduction section. Vaccination coverage, when did the vaccination start, which age group is targeted, these are missing in the introduction section. These inform the readers about the situation in the study area and support why the study was conducted.

Response: Thank you for this comment. We have revised the Introduction to provide insights into the national HBV vaccination programme in Somalia.

4) The limitations of the study are not sufficiently discussed (line 77-87).

4. The limitations of the study are not sufficiently discussed."

Response: Thank you for pointing out the need to expand our study limitations. In the revised manuscript, we have strengthened the Limitations section to provide a more transparent appraisal of the potential biases and interpretive constraints of our findings (474-481; 484-487).

5. I would recommend a linguistic revision of the manuscript by a native speaker as some parts do not read very well.

Response: We also appreciate your observation regarding the language and readability of the manuscript. The full text has now undergone thorough language editing, including review by a fluent English speaker.

Reviewer 2

This study looks at the Prevalence, risk factors, and viral load quantification of HBV infection among hospital patients: A cross-sectional investigation in Mogadishu, Somalia. It is well written but I have a few concerns for the authors to address.

1. 60…. Kindly clarify the body fluids through which transmission can occur since is not all body fluids

Response: Thank you for this important clarification. We have revised the sentence to specify that HBV transmission occurs primarily through infected blood, semen, vaginal secretions (line 59-60).

2. 214….. Kindly clarify what Dugsi/Madrasa is

Response: We have explained as footnote that Dugsi/Madrasa means islamic religious school (line 229).

3. 229…… 14.1% indicated they have chronic liver disease, what type of chronic liver disease. Does that mean some of the patients were aware of their hepatitis B status? Did you exclude patient who have screen for HBV already?

Response: Thank you once again. In our questionnaire, ‘chronic liver disease’ was captured as a self reported, broad clinical category and was not differentiated into specific aetiologies (HBV related cirrhosis, hepatitis C, or non viral liver disease), and we have now clarified this in the Methods and acknowledged it as a limitation. This variable therefore does not necessarily indicate that participants knew their hepatitis B status. We did not specifically exclude patients with a prior HBV test; however, individuals with a documented previous diagnosis were not systematically identified, and all eligible consenting outpatients were screened irrespective of past testing history. This clarification has been added to the Methods and Limitations sections (line 243-244).

4. 245 – 246…….In the methods, you subjected only the 31 samples to RT-PCR test, so you can claim those you did not do RT-PCR were negative. Therefore, this statement ‘’The RT-PCR results showed that 6 samples (2.2%) were confirmed positive for HBV DNA, indicating active viral replication, while 264 samples (97.8%) were negative’’ cannot be true. What about if some of the samples you did not do PCR for have occult hepatitis B infection

Response: Thank you one again for this important observation. We agree that our original phrasing could be misinterpreted as implying that all 264 HBsAg negative samples were PCR negative, although RT PCR was performed only on the 31 HBsAg positive specimens. We have revised the Results to report HBV DNA positivity as 6 of 31 HBsAg positive patients (2.2% of all participants), and we now explicitly state that HBV DNA testing was not done for HBsAg negative samples. In addition, we have added a limitation noting that occult HBV infection among HBsAg negative individuals could not be assessed and that the true burden of HBV infection may therefore be underestimated (line 261-267; line 478-481).

5. 371- 378 ….that among the infected individuals in our cohort, the vast majority (80.6%) were likely 371 inactive carriers with no detectable viral replication….. kindly replaced inactive carriers with a correct terminology

Response: Thank you for this important clarification. We agree that the term ‘inactive carriers’ is not strictly appropriate. We have therefore replaced this wording with a more accurate statement that reflects the data without implying a specific disease phase (line 387-403).

---

## [Decision Letter · Decision Letter 1]

28 Dec 2025

Prevalence, risk factors, and viral load quantification of HBV infection among hospital patients: A cross-sectional investigation in Mogadishu, Somalia

PONE-D-25-59057R1

Dear Dr. Garba,

We’re pleased to inform you that your manuscript has been judged scientifically suitable for publication and will be formally accepted for publication once it meets all outstanding technical requirements.

Kind regards,

Ashraf Elbahrawy

Academic Editor

PLOS One

Additional Editor Comments (optional):

Reviewers' comments:

Reviewer's Responses to Questions

**Comments to the Author**

Reviewer #1: All comments have been addressed

Reviewer #2: All comments have been addressed

2. Is the manuscript technically sound, and do the data support the conclusions?

Reviewer #1: Yes

Reviewer #2: Yes

3. Has the statistical analysis been performed appropriately and rigorously?

Reviewer #1: Yes

Reviewer #2: Yes

4. Have the authors made all data underlying the findings in their manuscript fully available?

Reviewer #1: Yes

Reviewer #2: Yes

5. Is the manuscript presented in an intelligible fashion and written in standard English?

Reviewer #1: Yes

Reviewer #2: Yes

Reviewer #1: Thank you to the Authors for addressing the comments and suggestions that I have made to the manuscript.

Reviewer #2: The authors have addressed all my concerns raised in my first review. the manuscript is look good for publication.

**Do you want your identity to be public for this peer review?** For information about this choice, including consent withdrawal, please see our Privacy Policy

Reviewer #1: Yes:  Fatemeh Farshadpour

Reviewer #2: Yes:  Dr. Amoako Duah

---

## [Editor Report · Acceptance letter]

PONE-D-25-59057R1

PLOS One

Dear Dr. Garba,

I'm pleased to inform you that your manuscript has been deemed suitable for publication in PLOS One. Congratulations! Your manuscript is now being handed over to our production team.

Kind regards,

on behalf of

Prof. Ashraf Elbahrawy

Academic Editor

PLOS One